# Gene Regulatory Network Controlling Flower Development in Spinach (*Spinacia oleracea* L.)

**DOI:** 10.3390/ijms25116127

**Published:** 2024-06-01

**Authors:** Yaying Ma, Wenhui Fu, Suyan Wan, Yikai Li, Haoming Mao, Ehsan Khalid, Wenping Zhang, Ray Ming

**Affiliations:** 1College of Agriculture, Fujian Agriculture and Forestry University, Fuzhou 350002, China; yaying123@yeah.net (Y.M.); fuwenhui0605@163.com (W.F.); 2Centre for Genomics and Biotechnology, Fujian Provincial Key Laboratory of Haixia Applied Plant Systems Biology, Key Laboratory of Genetics, Breeding and Multiple Utilization of Crops, Ministry of Education, Fujian Agriculture and Forestry University, Fuzhou 350002, China; wansuyan@163.com (S.W.); wsliyikai@163.com (Y.L.); mhaoming666@163.com (H.M.); ehsan.khalid33@gmail.com (E.K.); 3School of Breeding and Multiplication (Sanya Institute of Breeding and Multiplication), Hainan University, Sanya 572025, China

**Keywords:** *Spinacia oleracea* L., dioecious, monoecious, floral development, transcriptomics

## Abstract

Spinach (*Spinacia oleracea* L.) is a dioecious, diploid, wind-pollinated crop cultivated worldwide. Sex determination plays an important role in spinach breeding. Hence, this study aimed to understand the differences in sexual differentiation and floral organ development of dioecious flowers, as well as the differences in the regulatory mechanisms of floral organ development of dioecious and monoecious flowers. We compared transcriptional-level differences between different genders and identified differentially expressed genes (DEGs) related to spinach floral development, as well as sex-biased genes to investigate the flower development mechanisms in spinach. In this study, 9189 DEGs were identified among the different genders. DEG analysis showed the participation of four main transcription factor families, MIKC_MADS, MYB, NAC, and bHLH, in spinach flower development. In our key findings, abscisic acid (ABA) and gibberellic acid (GA) signal transduction pathways play major roles in male flower development, while auxin regulates both male and female flower development. By constructing a gene regulatory network (GRN) for floral organ development, core transcription factors (TFs) controlling organ initiation and growth were discovered. This analysis of the development of female, male, and monoecious flowers in spinach provides new insights into the molecular mechanisms of floral organ development and sexual differentiation in dioecious and monoecious plants in spinach.

## 1. Introduction

Plant reproductive systems exhibit remarkable diversity that is directly or indirectly related to their sexual reproduction. Among land plants, strict dioecy is rare, occurring in only 9–10% of species, which amounts to approximately 29,000 out of 300,000 estimated species. This distribution is notably uneven among flowering plants (angiosperms). Approximately 6% (15,600), spread across 987 genera and 175 families, adopt a dioecious reproductive strategy [1]. Angiosperm flowers typically consist of four primary organs; sepals, petals, stamens, and carpels. These components are organized on the receptacle, from the outermost sepals to the central carpels [2].

In recent years, substantial progress has been made in understanding sex chromosome evolution and sex-determining genes in angiosperms. Notably, the mechanism of sex determination mediated by TOPLESS-WIP1, influencing the expression of *CmCRC*, was elucidated in cucurbits. CmWIP1 and CmACS7 play pivotal roles in promoting the development of male and female flowers, respectively, with both genes being expressed in the carpel primordium. CmWIP1 inhibits the expression of *CmCRC* by recruiting TPL, thereby interfering with carpel determination and inducing male flower development. Conversely, the expression of *CmCRC* can promote carpel determination, and *CmACS7* is expressed after carpel determination, inducing female flower development [3,4]. On the other hand, sex determination in poplar (*Populus*) is regulated by a single gene (*ARR17*), which triggers androgynous development when expressed and leads to male abortion in the absence of its function [5]. In persimmon (*Diospyros kaki*), the *MeGI* gene encodes an autosomal transcription factor, and its expression is subject to regulation by the Y-encoded pseudogene (*OGI*), which encodes sRNA that acts upon *MeGI*. This regulation results in a reduction of *MeGI* expression, which leads to the development of male flowers, while female flowers exhibit a higher expression of *MeGI* [6]. Kiwifruit (*Actinidia deliciosa*) is morphologically monoecious and monogamous, but genetically dioecious. In A. deliciosa, sex determination is controlled by two genes (*SyGl* and *FrBy*). *SyGl* suppresses carpel development, and *FrBy* promotes stamen development [7].

Plants produce new organs through specialized regions called meristems, which consist of pools of stem cells capable of differentiating into specific tissue types. In particular, the inflorescence and floral meristems generate the reproductive structures that ultimately develop into flowers [8]. These reproductive organs may include stamens, carpels, or both [9]. Within the floral meristem (FM), the key transcription factors (MADS-box genes) for specifying floral organs are known as the ABCDE genes [10]. The well-known ABCDE model is divided into five different gene classes in Arabidopsis: class A *APETALA1* (*AP1*) and *APETALA2* (*AP2*) genes, sepals [11]; class A + class B *APETALA3* (*AP3*) and *PISTILLATA* (*PI*) genes, petals [12]; class B + class C *AGAMOUS* (*AG*) genes, stamens; class C genes, carpels [13]; class D *SHATTERPROOF* (*SHP*) and *SEEDSTICK* (*STK*) genes, ovule [14]; class E *SEPALLATA* (SEP) genes, all floral organs and ovules [14]. With the exception of *AP2*, the remaining genes in the ABCDE model belong to the MADS-box gene family. These ABCDE model genes have been considered as candidate sex determination genes in both monoecious and dioecious plant species [15].

Spinach (*Spinacia oleracea* L., 2n = 2x = 12) belongs to the family Chenopodiaceae. It is a widely cultivated annual or biennial crop, prized for its nutritious values as a green leafy vegetable. Plant flower types of spinach are divided into the following four categories: dioecy, monoecy, gynomonoecy, and androdioecy [16]. Dioecious spinach has an XY sex determination system, where XX individuals are female plants and XY individuals are male plants [14,16]. Notably, spinach is among the species with viable YY genotype plants, indicating that the sex chromosomes of spinach have evolved to the second stage of sex chromosome evolution [16,17]. Moreover, some varieties and strains of spinach that do not rely on the XY sex chromosome system to regulate sex expression are capable of producing monoecious plants that bear distinct male and female flowers and are controlled by a single locus (M) [18]. Genetic analysis has revealed that Y and M are epistatic to M and X, respectively: XXmm represents female, while XYMM, XYMm, and XYmm are male, and XXMM and XXMm produce a monoecious phenotype [19,20,21]. Northern blot has shown that *SpAP3* is strongly expressed in male flowers compared to female flowers; in contrast, *SpPI* is expressed in the early stages of male flower development and is absent in all stages of female flower development [22].

Virus-induced gene silencing (VIGS) of class B genes (*SpPI* and *SpAP3*) results in a homeotic transformation from stamens to carpels, affecting the number of perianth and the existence of the fourth whorl [23]. Sequence analysis of *SpPI* and *SpAP3* shows no allelic differences between males and females, implying that sex-specific development is controlled through the regulation of trans-acting factors expressed by class B genes [23]. The C class gene *SpAG* is expressed early in the entire floral primordium before the emergence of male and female floral organ primordia. In later stages of flower development, *SpAG* is specifically expressed in male microsporangium cells and female nuclear cells [24]. A previous study has identified *SPGAI* (GIBBERELLIC ACID INSENSITIVE), a member of the DELLA family of the transcription factor, as the key feminizing factor in spinach. *SPGAI* plays a pivotal role in initiating the development of unisexual organs. It is important to note that the feminization pathway is regulated by *SPGAI* being epistatic to the masculinization pathway [25].

To identify the genes involved in the development of spinach flowers, we compared the structural differences between female and male flowers in dioecious plants from the floral differentiation to mature stages, as well as the structural differences in flowers between dioecious and monoecious flowers. Additionally, we identified DEGs in different developmental stages of female and male flowers, respectively, and DEGs of female and male flowers at the same developmental stage through transcriptome data analysis. We aimed to analyze the flower developmental processes in spinach, with a particular focus on the determination of floral organ identity, morphogenesis, and maturation. This study will enhance our understanding of differential gene expressions during the development of female and male spinach flowers. We sought to identify genes exhibiting differential expression in dioecious and monoecious inflorescences with unisexual flowers, or those displaying gender-biased expression. Finally, through comparative analysis of the expression patterns of flower development genes in other flowering plants, we determined whether these patterns in spinach are consistent with or divergent from those observed in other species.

## 2. Results

### 2.1. Dynamic Transcriptomic Profiles during Female and Male Flower Development in Dioecious Spinach

In dioecious spinach, male and female flowers first differentiate at the beginning of the sepal primordia before the formation of reproductive primordia [24] (Figure 1A). Female flowers have approximately four white stigmas and sepals, while male flowers have approximately four yellow anthers at the end of filaments. There were differences in differentially expressed genes among the female and male flower development stages. The overall expression in spinach female flowers was relatively uniform, while the expression in male flowers fluctuated slightly (Figure 1B). The number of genes with fragments per kilobase of transcript per million mapped reads (FPKM) values between 5 and 50 in male flowers was greater than those in female flowers, while the number of genes with FPKM values greater than 100 in female flowers was more than those in male flowers. Based on the four comparisons of S2 vs. S3, S3 vs. S4, S4 vs. S5, and S5 vs. S6 in female and male flower development, the number of DEGs in male flowers was 9.42 times that in female flowers (Figure 1C).

### 2.2. Functional Categories of DEGs Involved in Dioecious Spinach Flower Development

To further investigate the supposed functions of DEGs during dioecious spinach flower development, Kyoto Encyclopedia of Genes and Genomes (KEGG) and Gene ontology (GO) analyses were performed. The following 8, 9, 10, 4, 29, 18, and 15 KEGG pathways were significantly enriched in the F3 vs. F4, F4 vs. F5, F5 vs. F6, MS2 vs. MS3, MS3 vs. MS4, MS4 vs. MS5, and MS5 vs. MS6 comparison sets (Appendix A). In these pathways, ‘biosynthesis of secondary metabolites’, ‘phenylpropanoid biosynthesis’, ‘alpha-linolenic acid metabolism’, ‘tryptophan metabolism’, and ‘metabolic pathways’ were all significantly enriched from F3 to F6 in the female flowers. ‘Pentose and glucuronate inter-conversions pathway’ and ‘amino sugar and nucleotide sugar metabolism pathway’ were significantly enriched from M2 to M6 in the male flowers. All of the KEGG pathways associated with growth and energy costs showed some distinctions in the stages of spinach flower development.

Significant enrichment analysis of GO functions was performed on DEGs during the development of female and male flowers. There were six O-methyltransferase activity terms and one transcription factor activity term in the female flowers (Appendix A). While, 4 glucosyltransferase activity terms, one transcription factor activity term, and one pectinesterase activity term in male flowers (Appendix A). In biological processes, female and male flowers are very different. Female flowers have two terms related to meristem development, and male flowers have three terms related to pollen development (Appendix A). In addition, based on the GO annotation results of the DEGs, there are 235 biological processes in female and male flowers related to stem cell, reproduction, meristem, morphogenesis, floral organ development, axis, and polarity (Appendix A). In addition, GO entries involved in hormone synthesis or in response to hormone signal transduction were also detected. During the flower meristem stage, ABA, auxins, ytokinin (CK), ethylene (ET), and GA are mainly involved. Totals of six hormones (ABA, auxin, brassinosteroids (BRs), ET, GA, and jasmonic acid (JA)) and three hormones (ABA, auxin, and salicylic acid (SA)) participate in male and female flower development, respectively (Figure 2A). Thus, the activation of plant hormone signaling at the bud flower and full-bloom stages might favor spinach flower development.

### 2.3. Identification of Transcription Factors in Dioecious Spinach Flower Development

Among the 1268 TFs identified genome-wide in this study, differentially expressed TFs exhibited varying expression patterns in female and male flowers during flower development (Appendix A). A total of 548 differentially expressed TFs were identified in female and male flowers (FS2–FS6, MS2–MS6), with bHLH (112), NAC (106), FAR1 (89), MYB (77), ERF (67), B3 (64), and C2H2 (64) families being the most abundant (Appendix A). Additionally, 758 DEGs were identified from 44 TF families by comparing the floral meristem formation and differentiation of female and male flowers (FS2 vs. MS2, FS2 vs. MS3), carpel and stamen primordia formation and differentiation (FS3 vs. MS4), and pistil and anther development (FS4 vs. MS5, FS5 vs. MS5) (Figure 2B). The most abundant TF gene family was bHLH (70), followed by NAC (65), MYB (61), MIKC_MADS (50), and ERF (50). The number of the same TF was significantly different between the two genders. NAC and MYB were mainly expressed during stamen development, while ERF, WRKY, HD-ZIP, and GRAS were predominantly expressed during carpel development. GRF and YABBY were specifically expressed in carpel. M-type_MADS and ARR-B were associated with stamen development. The expression levels of the MIKCc and ABCDE model genes showed varying degrees of differential expression during the development of male and female flowers (Figure 2C).

Meanwhile, 18 TFs were detected only in the female flowers and 54 TFs in the male flowers (Figure 3A). For instance, *SOV2g004450.1* (*ERF*) exhibited high expression levels in the female flowers and down-regulation at the mature stage. The expression levels of *SOV4g032440.1* (*AGL*), *SOV1g045060.1* (*YABBY*), and *SOV3g036900.1* (*bHLH*) started to increase during later stages, which might have contributed to the morphogenesis and maturation of the female flowers. *SOV3g000300.1* (*C2H2*) was only expressed in male FM, and two PI (*SOV2g002560.1*, *SOV2g002570.1*) were specifically expressed during male flower development. *NAC* (4), *MYB* (4), *LBD* (2), *C2H2* (2), and *MYB_related* (2) were highly expressed specifically during the differentiation period of stamen primordia. *SOV4g039600.1* (*NAC*), *SOV6g019470.1* (*ZF-HD*), *SOV4g047880.1* (*ERF*), and *SOV5g007750.1* (*LBD*) were up-regulated at a later stage in the male flowers. Three and eight TFs were detected and expressed at MS5 and MS6, respectively.

Based on the regulatory relationships among the 15 investigated floral regulators in Arabidopsis, including factors controlling floral transition (FLC, FLM, SVP, and SOC1), FM/organ-identified TFs (LFY, AP1, AP2, AP3, PI, SEP3, and AG), and regulators of floral organ morphogenesis (BLR, JAG, ETT, and RGA) [26], we classified potential target DEGs of these regulators into protein-coding gene loci. Totals of 14.5% (132 out of 913) and 14.8% (1039 out of 6998) DEGs were identified as potentially regulated by one floral regulator during female and male flower development, respectively (Appendix A). During the entire flower development period, including the early, full, and late stages, the high-frequency differential genes were regulated by the flower regulatory factors, as illustrated in Figure 3B. In addition to the above-mentioned floral regulators, we constructed a few of the GRNs between the TFs and genes in the meristem tissue of female and male organ development among the DEGs in spinach (Appendix A).

In summary, our results show that the TFs between the females and males were observed to be quite different. The TFs during the FM stage to the mature stage in spinach dioecious plants are expected to be involved in sex differentiation and probably associated with floral organ development and reproduction.

### 2.4. Transcriptome Analysis of Spinach Axillary Buds

To determined the sex-biased DEGs in spinach, we confirmed 2,356 DEGs (Padj < 0.01, |log2FoldChange | > 2) during the early development stages of female, male, and monoecious (XXMM) spinach axillary buds (including stage 1 to stage 4) (Figure 4A). A total of 476 sex-biased genes were identified (Figure 4B), with male-biased genes being the most abundant, followed by female-biased genes, and monoecious-biased genes. Male-biased genes were evenly distributed across spinach chromosomes, with the largest distribution on chromosome 4; female-biased genes were also most abundant on chromosome 4. Monoecious-biased genes showed a distribution similar to male-biased genes, but were most abundant on chromosome 1. Other than this, four (YABBY, trihelix, GATA, and ZF-HD), five (C2H2, bZIP, M-type-MADS, NZZ/SPL, and NAC), and two (Co-like, and Dof) TF families among the DEGs were specifically expressed in the female, male, and monoecious axillary buds, respectively. In addition, some TFs were expressed in more than two genotypes (Figure 4C). Among the sex-biased DEGs, 2, 11, and 3 genes involved in reproductive organ development were specifically expressed in the female, male, and monoecious axillary buds, respectively (Figure 4D).

### 2.5. Comparing the Differences in the Mature Stage of Dioceious and Monoecious Flowers in Spinach Flower Development

Heterozygous genotypes of monoecious plants (XXMm) exhibit both female and male flowers (Figure 5A). By comparing the transcriptome data at the maturity stage of dioecious (female, male) and monoecious flowers (Figure 5B), a total of 5518 DEGs (Padj < 0.01, |log2FoldChange | > 2) were identified. The FS6 vs. MS6 comparison group had more DEGs (Figure 5C), and 317 DEGs were differentially expressed among the three genders. These 317 DEGs were enriched for various TFs, with bHLH, ERF, MYB, and C2H2 being relatively abundant (Figure 5D). There were six DEGs in the MYB and NAC families, respectively, that were specifically expressed in mature male flowers (Figure 5E). TCP and YABBY TFs were not detected in male flowers and were present only in female and monoecious flowers.

To further investigate the regulation mechanisms of flowers at mature stages of different genders (female, male, and monoecious), 19 KEGG pathways were significantly enriched (Appendix A). ‘Pentose and glucuronate interconversions’ (66), ‘Plant-pathogen interaction’ (67), ‘Starch and sucrose metabolism’ (70), ‘Phenylpropanoid biosynthesis’ (60), ‘Metabolic pathways’ (546), ‘Plant hormone signal transduction’ (63), and ‘Biosynthesis of secondary metabolites’ (302) had the highest numbers of DEGs associated with flower development at the maturity stage of the three genders of spinach.

By performing GRN analysis on the DEGs of female and male flowers at the maturity stage, we found that 54 TFs were significantly enriched and regulated female flowers and monoecious flowers, which indicates that these TFs might regulate carpel maturation. Nine TFs regulated the development of male and monoecious flowers at the mature stage, indicating that they are related to the development of stamens at the mature stage. Eight and two TFs specifically regulated male and monoecious flowers, respectively, indicating that these genes may be involved in different mechanisms to regulate stamen development (Appendix A).

### 2.6. Validation of Selected Gene Expressions

The transcriptome data from female and male flowers of dioecious plants have been validated [27,28]. The expression levels of four genes associated with the axillary and maturity stages of different genders of spinach were validated by reverse transcription–polymerase chain reaction (RT-qPCR), and the quantitative analysis results were compared with the expression trends of the FPKM values (Figure 6A,B). The RT-qPCR results show that *SpCRC* and *SpTFIIB* were specifically highly expressed in the axillary buds of female and monoecious plants. *SpPI* and *SpAP3* were expressed in the axillary buds of male and monoecious plants, particularly showing high expression in the axillary buds of male plants. *SpAGL6* and *SpSTK* exhibited high expression in the mature flowers of female and monoecious plants, while *SpAP1* showed high expression in the mature flowers of female plants. *SpPI* was highly expressed in the mature flowers of male and monoecious plants. These results suggest that the expression patterns of RT-qPCR are consistent with those of FPKM, demonstrating the reliability of the transcriptome data. Furthermore, we examined the spatial expression patterns of *SpTFIIB* using in situ hybridization. *SpTFIIB* showed a specific expression pattern in monoecious axillary buds (Figure 6C), consistent with the transcriptome and RT-qPCR results.

## 3. Discussion

### 3.1. Comparing the Global RNA-Seq Analysis of Dioecious Spinach Flower

In spinach, there has been limited exploration of the gene expression changes throughout its flower development. Therefore, we divided the experimental materials of spinach flowers into three periods: organ identity determination, morphogenesis, and maturation, with the aim to further investigate the genes that regulate spinach flower development through detailed analysis. Both KEGG and GO enrichment analysis showed that there were significant differences in the development of female and male flowers in dioecious spinach (Appendix A). TFs play pivotal roles in various aspects of plant growth and development [29]. A considerable number of TFs identified in our study seemed to be associated with roles in spinach flower identity, determination, and development (Figure 7). Previous analysis has shown that auxin and GA are common key factors in regulating the development of male and female flowers [30]. On this basis, we also detected that ABA might be involved in regulating flower development (Figure 2A). In summary, our results suggest that DEGs with diverse molecular functions participate in distinct biological pathways, and KEGG pathways contribute to the formation of spinach flowers with the involvement of various TFs.

### 3.2. Determination of Floral Meristem Formation in Spinach

Several key genes associated with floral meristem identity (FMI) were examined in the context of spinach flower development. *LEAFY* (*LFY*), *AP1*, *AP2*, and *CAULIFLOWER* (*CAL*) [31,32,33], known as FMI genes, were found to be highly expressed in the floral meristems (FMs) of both female and male flowers of spinach (Appendix A). LFY, in conjunction with *UNUSUAL FLORAL ORGANS* (*UFO*), regulates *AP1* and *SEP* genes [34,35]. Additionally, *SUPPRESSOR OF OVEREXPRESSION OF CO 1* (*SOC1*), working with *AGL24*, was highly expressed in the early stages of female and male flower development, promoting flowering and inflorescence meristem identity [36].

Comparing the spinach floral meristem development stages, DEGs were identified expressed in both female and male flowers. For instance, *EXPA9* was detected at the initiation stage of both female (up-regulated) and male (down-regulated) flowers as DELLA-down genes in the young flower buds [37]. *YAB1* and *CcYAB5*, expressed in various plant organs [38], exhibited different expression patterns compared to *CRABS CLAW* (*CRC*), which was expressed throughout female and male flower development. The transcripts of *LFY* are substantially reduced in the shoot apices of *pny pnf* double mutants after floral induction. Double mutants (*pny pnf*) do not produce flowers, while *SpPNF* was expressed in the early stages (stages 2–3) [39], indicating that *SpPNF* is involved in the meristem identity process. In Arabidopsis, *ROXY2* can regulate anther development [40], whereas in spinach, *SpROXY2* is expressed in the early flower development stages of spinach (stages 2–4). This indicates that these genes might be more significantly correlated with the maintenance of spinach FM development.

Phyto-hormones, particularly auxin signaling, are crucial for triggering and maintaining the FM, female and male primordia, and development. Auxin plays a pivotal role in FM initiation and gynoecium development [41]. *PIN-FORMED 1* (*PIN1*), involved in the auxin pathway, appears to be the most important for reproductive development and can be activated by ABCB protein [42,43,44]. SlPIN1a plays a role in meristem maturation and floral organ specification [45], and *SpPIN* is expressed in the FM (Appendix A).

Auxin signaling involves various factors, including AUXIN RESPONSE FACTORS (ARF) and IAA/AUX [46]. *ARF3* has a dynamic role in patterning by acting in specific cells within FMs and reproductive organs [47]. *SpARF*, *SpAUX1*, *SpIAA*, *SpGH3*, and *SpSAUR* act synergistically during female and male flower development. PYR/PYLs are abscisic acid-receptors that function at the apex of a negative regulatory pathway to control abscisic acid (ABA) signaling by inhibiting type 2C protein phosphatases (PP2Cs) [48]. In this study, *PP2C* was also repressed by PYL throughout the female and male flower development (Appendix A). ABA can be involved in the regulation of circadian rhythms (ko04712) to regulate flower induction and transition in plants [49]. *SpCOP1*, *SpFKF1*, *SpCRY1*, *SpPHYB*, *SpTOC1*, *SpPRR9*, *SpLHY*, and *SpCO* were involved in the circadian rhythms expressed throughout the early flower development stages to later stages in spinach (Appendix A). *CYCD3*, related to cell division, showed high expression levels in the FM of females and males (FS2–FS3, MS3) (Appendix A), and its expression level gradually decreased in the later stages. This suggests that BR-related genes may be involved in the determination of the floral meristem. CK plays a key role in the regulation of cell formation and differentiation of FM [50], affecting flowering-related gene expression levels in apple [51]. Our results indicate that hormone-related genes are involved in the regulation of spinach flowers at the FM stage and later stages.

### 3.3. Determination of the Identities of Floral Organs in Spinach

The flower primordia undergoes rapid and coordinated bursts of cell expansion and division in three dimensions, generating a concentric group of cells as an almost spherical structure from which all floral tissues derive after the formation of an FM [52]. Soon after FM specification, it subdivides into four distinct regions. Each region gives rise to primordia of different whorls, sepals, petals, stamens, and carpels [52,53]. In dioecious spinach, before reproductive primordia formation, male and female flowers first differentiate at the initiation of the first whorl of primordia. Males produce four sepal primordia, females only two [24], and there are no petals in spinach.

MADS-box genes are key regulators to control the identity and patterning of floral organs during development. *AG* is co-regulated by *WUSCHEL* (*WUS*) [54,55] and expressed in stamen and carpel primordia. This directly induces *SPL*/*NZZ* [56]. *SEP* (E class) genes interact with various combinations of A, B, and C class floral organ identity genes to specify the identities of sepals, petals, stamens, and carpels [35]. *SEP1*/*AGL2*, *SEP3*, and *SEP4*, as E-class genes, are regulated by *LFY* for organ identity specification [35,57] and were expressed during the development of female and male flowers (Figure 2C) [58]. *PI* and *AP3* are first induced by *LFY*/*UFO* in response to flowering signals [59]. Two paralogs of SpPI are expressed in male flowers specifically [23]. *SpAP3* regulates stamen identity, and is expressed during dioecious female and male flower development. B-class genes are crucial in petal development [60], but there are no petals in spinach. It has been speculated that *SpAP3* may have an additional function in regulating female development. *AGL24* is a mobile mRNA whose movement is necessary and sufficient to specify floral organ identity in Arabidopsis [61] and was expressed in FS2 and MS2. The above results indicate that MADS-box gene family members are involved in determining the identity of spinach sepals, stamens, and carpels, which is consistent with previous results.

### 3.4. Morphogenesis of Floral Organs in Spinach

A floral organ undergoes morphogenesis to change its size and shape over time after initiation. While different floral organ types (e.g., sepals, petals, stamens, and carpels) typically follow distinct developmental trajectories, the underlying mechanisms are similar, involving the proliferation, expansion, and differentiation of cells, as well as the establishment of adaxial–abaxial, proximal–distal, and lateral–medial tissue polarities [2]. YABBY family members (*YAB1-3*) determine abaxial fate, while adaxial cell fate is determined together with *REVOLUTA* (*REV*), *PHABULOSA* (*PHB*), and *JAGGED* (*JAG*), which together form a module that regulates adaxial–abaxial primordia polarity [62,63]. *CRABS CLAW* (*CRC*), a member of the YABBY family, is directly regulated by *AG* and indirectly regulated by *LFY* and *UFO* to develop nectary and carpel. *SpCRC* is only expressed in carpel, similar to *CmCRC* in cucurbits [4]. *SpREV* and *SpPHB* showed high expression levels during all flower development stages (Appendix A). *YUCCA* (*YUC*) and *PIN* control the biosynthesis and polar transport of auxin. SPL may regulate auxin homeostasis by repressing the transcription of *YUC2* and *YUC6* and participate in lateral organ morphogenesis. It has been suggested that they could also be involved in the adaxial–abaxial domain establishment [64]. These results suggest that the above DEGs might determine the morphology of dioecious spinach lateral organs.

### 3.5. Maturation of Floral Organs in Spinach

When the floral organs are differentiated, they enter into the mature stage. The style and stigma of the female spinach flower receive pollen, and the ovules expand. The pollen of male flowers ruptures to facilitate the release of pollen grains. *KCS6*, encoding 3-ketoacylCoA-synthase in the fatty acid elongation pathway, is a major condensing enzyme involved in stem wax and pollen coat lipid biosynthesis, highly expressed in the tapetum of anthers near maturity [65,66]. Two *SpKCS* (*SOV4g010570.1* and *SOV1g032730.1*) were detected and expressed in males specifically (Appendix A). Cutin, suberine, and wax biosynthesis pathways can participate in the development of the exine of microspores in the tapetum and anther expansion [67], and many related genes were enriched in MS4 and MS5. In flower development, amino acids play a crucial role in enzyme synthesis and osmotic regulation, providing nitrogen and energy for pollen and ovule maturation [68]. In spinach, the amino acid metabolism pathway was mainly enriched in male flower development, while the amino acid degradation pathway was mainly enriched in female flower development (Appendix A). The energy and carbon sources consumed during flower development may come from sucrose and starch metabolism generated by photosynthesis and carbon metabolism [69,70], while amino acids are used more as structural components to maintain the proper state and shape of the flower [71]. Among the DEGs, some genes encoding pectinesterase were detected (Appendix A). PECTIN METHYLESTERASE48 (PME48) affects the mechanical properties of the inner wall of pollen grains during maturation in Arabidopsis, thereby affecting pollen grain germination [72], suggesting that pollen-specific PMEs in spinach male flowers contribute to pollen development.

ABCDE model genes also contribute to later development stages of spinach flowers. *SPL* is directly induced by *AG* and is expressed during micro- and megasporogenesis to be activated at later stages of stamen development and expressed in the sporogenous tissue of anthers and ovules [73]. *SpSPL* contributes to female and male flowers in spinach (Appendix A). In addition to the B-class genes that regulate stamens, there are some *AGAMOUS-LIKE* (*AGL*) genes also are involved in regulating stamens. *AGL30*, *AGL65*, and *AGL104* regulate pollen activity and are required for pollen germination and tube growth [74]. D-class genes are proposed to specify ovule identity with *AG*, but only *STK* was detected in spinach [14,75]. *STK* can interact with E-class proteins (SEP1/3/4) to regulate ovule identity and differentiation [14].

Phyto-hormones play an important role in late-developmental organs. JA is involved in filament elongation, anther dehiscence, and pollen production [76]. Mutants deficient in JA synthesis result in male sterility [77]. Auxin signaling accelerates the maturation and development of pollen, anthers, and ovules [78]. GA regulates cell elongation in developing filaments and anther cell differentiation from microspores to mature pollen grains [79]. Some TFs respond to one or more classes of plant hormones. The MYB family was the most enriched in phyto-hormone-responsive genes (GA, SA, ABA, and JA) [80,81,82,83,84] and plays roles in sperm cell specification [85], stamen and pollen maturation [86], and anther tapetum and stigma papillae development. NAC TFs play important regulatory roles in various parts of floral organs at different stages of development. Particularly during the anther development process, they can interact with MYB103 to regulate pollen development [87], work with other TFs, such as MYB99, to regulate the formation of pollen walls, affecting pollen development [88], and can interact with MYB26 to regulate pollen dehiscence [89]. LBD (microspore-specific proteins), AtLBD27/SIDECAR POLLEN (SCP), and AtLBD10 contribute to male gametophyte development [90,91]. Five *LBDs* were detected and expressed in the lateral organ development of male spinach (Figure 3A). bHLH, as a BR-responsive TF family, contributes to stigmatic cell development in Arabidopsis [92]. Most of the bHLH TFs were expressed highly during the maturation stage of the female flowers and monoecious plants (Figure 5E). Additionally, SPL/NZZ may act as an adapter to recruit TOPLESS (TPL) to TCP target genes, thus preventing the overexpression of these genes and disrupting ovule development [93]. A total of 13 *TCPs* were detected in spinach and expressed highly in female and monoecious mature flowers, while *SpTPL* was also highly expressed throughout the development of female flowers. This suggests that *TCPs* might be regulated by SPL and *TPL* to regulate ovule development in spinach.

Taken together, the expressions of TFs and the presence of different flower development regulators, including various plant hormones as well as their interactions, demonstrate the complexity of the regulatory network involved in flower development.

### 3.6. Identification of Sex-Biased Candidate Genes

As previously reported studies, *NRT1* in the Y-specific region might regulate stamen and carpel initiation through two independent pathways (*NRT1-JA*/*GA-PI* and *NRT1-PI-CRC*/*KNU*/*WUS*) [28]. In this study, we mainly focused on genes with sex-biased expression outside the sex-determining region in the sex chromosomes. Furthermore, PI acts as the initial gene for stamen development [23] and is also expressed in monoecious flowers in spinach, suggesting that it might be regulated by autosomal genes other than male-specific genes in the Y chromosome. *CRC* might be the genes responsible for carpel initiation, which was reported in spinach [28] and is consistent with the expression of *CmCRC* in melon [94]. In dioecious spinach, *SpCRC* was expressed in the early stages of female flower development and was not expressed during the entire period of male flower development (Figure 2C).

In addition to those organ-specific genes in the MADS-box family, we detected other TFs that were specifically expressed in different genders. LBD, NAC, and MYB TFs regulate female and male reproductive organ development [87,90,91,95,96]. Several of the TFs discussed above were specifically expressed in the male or female gametophyte stages (Figure 3A). There were 3, 4, 6, 6, and 11 genes belonging to the LBD, B3, MYB, NAC, and bHLH families, respectively, that were significantly expressed in the monoecious mature flowers (Figure 5E). These genes might synergistically regulate the maturation of stamens and carpels. At the same time, we found that *TFIIB* was specifically expressed in monoecious flowers (during the early stages) (Figure 4D). AtTFIIB1 plays vital roles in endosperm and pollen tube growth, guidance, and reception [97]. RT-qPCR and in situ hybridization experiments showed that *SpTFIIB* was specifically expressed in monoecious axillary buds (Figure 6A,C). Future studies will perform transgenic function experiments to verify the function of *SpTFIIB*.

### 3.7. Underlying Functional Gene Regulatory Modules during Dioecious Spinach Flower Development

By integrating all transcriptome analyses conducted in this study, we propose a model potentially regulating floral development and sexual differentiation (Figure 8). We have identified key genes involved in the stages from spinach flower differentiation to the maturation of pistils and stamens, as well as candidate genes specifically expressed in the anthers and ovules during the maturation stage.

## 4. Materials and Methods

### 4.1. Plant Material and RNA Extraction

Monoecious spinach cultivar PI 604782 (accession number SPI 12/79) was obtained from the USDA. II9A0073 cultivar was provided by the Chinese Academy of Agricultural Sciences (CAAS). Spinach plants were grown in the greenhouse under 16 h light/8 h dark at 22 ℃ at the Fujian Agriculture and Forestry University (Fujian, China). The periods shown in Figure 1A correspond to a previous study [24], where female and male flowers were compared from the floral meristem stage to the floral maturity stage (stage 2 to stage 6) and were sampled as shown in Figure 1A. The samples taken at stage 2 include those from stage 1. In addition, mature stage samples of monoecious flowers were sampled, containing both female and male flowers. Axillary bud (AB) samples of monoecious and dioecious flowers were sampled, named ‘MoAB’, ‘FeAB’, and ‘MAB’ (Figure 4A). Samples from each period were observed under a stereo microscope (LEICA, DFC550, Wetzlar, Germany) and photographed. Three replicates of each sample were collected, immediately frozen in liquid nitrogen, and stored at −80 °C before use. The total RNA were extracted from the spinach samples prepared for transcriptome sequencing and RT-qPCR using the RNeasy Plant Mini Kit (QIAGEN, Hilden, Germany). The integrity and quality of RNA were tested by 1.5% agar gel electrophoresis and NanoDrop 2000 (Thermo Fisher Scientific, Waltham, MA, USA).

### 4.2. RNA-Seq Library Preparation, Sequencing, and Analysis

An NEBNext^®^ Ultra^™^ RNA Library Prep Kit for Illumina^®^ (https://www.neb.com/en-us/-/media/nebus/files/manuals/manuale7770_e7775.pdf?rev=c1793efa50d6407e80abdd40dd5646cc&hash=9B88177FBDFB8EA952D71226A6BF4ABD (accessed on 1 April 2022)) was used to construct the RNA-Seq libraries. The sample library concentrations were evaluated using the Qubit^®^ 2.0 fluorimeter with the Qubit dsDNA HS Assay Kit before sending them to Novogene (Beijing, China) for sequencing on the Illumina NovaSeq 6000 System. We conducted quantitative analysis of the RNA-Seq data using Trinity Transcript Quantification. For the detailed methods, please refer to the website: https://github.com/trinityrnaseq/trinityranseq/wiki/Trinity-Transcript-Quantifcation (accessed on 1 June 2022). The default comparison software Bowtie2 [98] (https://bowtie-bio.sourceforge.net/bowtie2 (accessed on 1 June 2022)) was used to quickly and precisely compare the clean reads with the reference genome (a total of 28,964 protein-coding genes) [99], and then RSEM [100] (https://github.com/deweylab/RSEM (accessed on 1 June 2022)) was used to calculate the FPKM values, and genes with an adjusted *p*-value < 0.01 (|log2(fold change)| ≥ 1), found using DESeq2 [101], were considered differentially expressed genes (DEGs). All heatmaps were drawn using the R package pheatmap, and the expression values were scaled by log2(FPKM + 1).

### 4.3. Function Enrichment Analysis

KEGG pathway enrichment analysis [102] and GO enrichment analysis of the DEGs was performed using the EggNOG (v5.0) Database (http://eggnog5.embl.de/#/app/home accessed on 1 January 2024) and visualized by the OmicShare online tool (https://www.omicshare.com/ accessed on 1 January 2024). The TF families were classified by plant transcription factor database PlantTFDB 4.0.

### 4.4. Gene Co-Expression Construction

Mutual rank (MR) analysis was used to identify significant co-expression relationships among DEGs, and the detailed analysis methods described by Liao [103] were used in this study. The final network was visualized using Cytoscape version 3.6.1.

### 4.5. RT-qPCR Analysis

The cDNA was synthesized using a Prime Script RT reagent kit (TaKaRa, Kyoto, Japan). The cDNA was diluted with nuclease-free water, and its final concentration was 200 ng per microliter. The selected gene primers for RT-qPCR were designed with IDT (https://sg.idtdna.com/pages (accessed on 1 January 2024)) (Appendix A). The RT-qPCR was conducted in a 12.5 μL reaction system with the following components: 6.25 μL *SYBR Premix Ex Taq*II, 0.5 μL 0.4l M forward primer, 0.5 μL 0.4 L M reverse primer, 0.5 μL diluted cDNA, and 4.75 μL ddH2O. RT-qPCR was conducted on the Bio-Rad CFX96 Real Time PCR System (Bio-Rad, Hercules, USA). The procedure was as follows: 95 °C for 3 min, 95 °C for 40 cycles of 10 s, and 55 °C for 30 s. The melting curve was generated from 65 to 95 °C, and the temperature was increased by 0.5 °C every 5 s. Axillary buds and stage 6 RNA samples of monoecious and dioecious flowers were obtained as described in Section 4.1. Each gene was detected in three biological replications and three technical replications. *Actin11* gene was used as a reference gene and as an internal control. The relative expression level was calculated by the 2^−ΔΔCT^ method. Column charts and line charts were visualized using Excel.

### 4.6. In Situ Hybridization

A 217-bp fragment from the 3′ end of the TFIIB cDNA sequence was subcloned into pTA2 Vector (TOYOBO, Osaka, Japan) for the production of both sense and antisense RNA probes. The primers of sequences are shown in Appendix A.

After collecting the axillary buds with different gender expressions from the spinach (Figure 4A), they were immediately immersed in a plant-fixative 4% paraformaldehyde solution for 24 h. The samples were dehydrated using a graded ethanol series (50%, 70%, 90%, 95%, and 100%, 60 min for each step). Xylene was then gradually used to replace the ethanol (in ratios of xylene/ethanol of 1:3, 1:1, and 3:1), and finally, the samples were immersed in pure xylene. Paraffin was gradually added to the solution containing the samples, eventually replacing it with pure paraffin. Once fully infiltrated with paraffin, the samples were embedded in paraffin blocks, and continuous sections were cut using a Leica microtome (Leica, Wetzlar, Germany). All these operations were conducted in containers free of RNase. In situ hybridization experiments were performed as described by Sather [24]. After hybridization, the slides were observed and subsequently photographed using a microscope (OLYMPUS, Tokyo, Japan).

### 4.7. Quantification and Statistical Analysis

Using normalized RNA-Seq data (FPKM values) and log2 fold change to calculate the differential expression levels, the P-values were calculated with Student’s *t*-test. *p*-values < 0.05 are considered significant. K-means was used to cluster the gene expression data (Figure 2C, Figure 4D, and Figure 5E).

## 5. Conclusions

Transcriptome analysis of dioecious spinach female and male flowers from the floral meristem to mature stages was performed for the first time, and we also analyzed the transcriptomic data of monoecious flowers. Our results provide an informational framework of TFs, genes, and hormones, illuminating the complexity of this development process. They contribute to the understanding of the molecular regulatory mechanisms underlying female/male plant development in spinach, although further experiments are needed to validate the precise roles and interactions among the candidate genes. Overall, this study provides a new theoretical foundation and research avenue for investigating the sex differentiation and reproductive development of spinach.

## Figures and Tables

**Figure 1 ijms-25-06127-f001:**
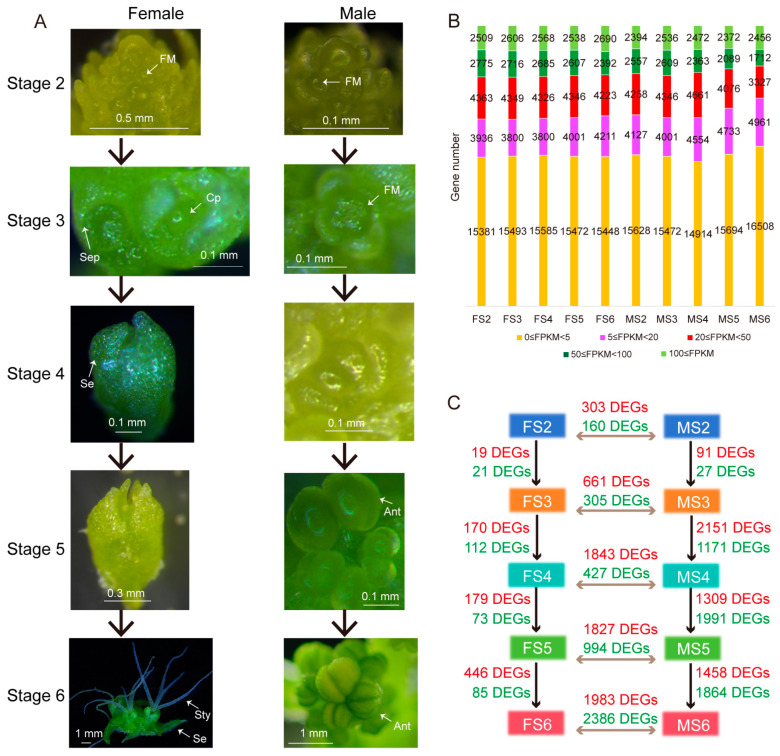
Global transcriptome changes in five developmental stages of spinach female and male flowers. (**A**) The morphology of spinach male and female flowers in the five flowering stages. FM: floral meristem; Cp: carpel primordium; Sep: sepal primordia; Se: sepals; Ant: anther; Sty: style. (**B**) Transcriptional dynamics during spinach female and male flower development showing the number of DEGs at five stages. F: female; M: male; S: stage. (**C**) The number of up-regulated (red font) and down-regulated (green font) DEGs in female and male flower development.

**Figure 2 ijms-25-06127-f002:**
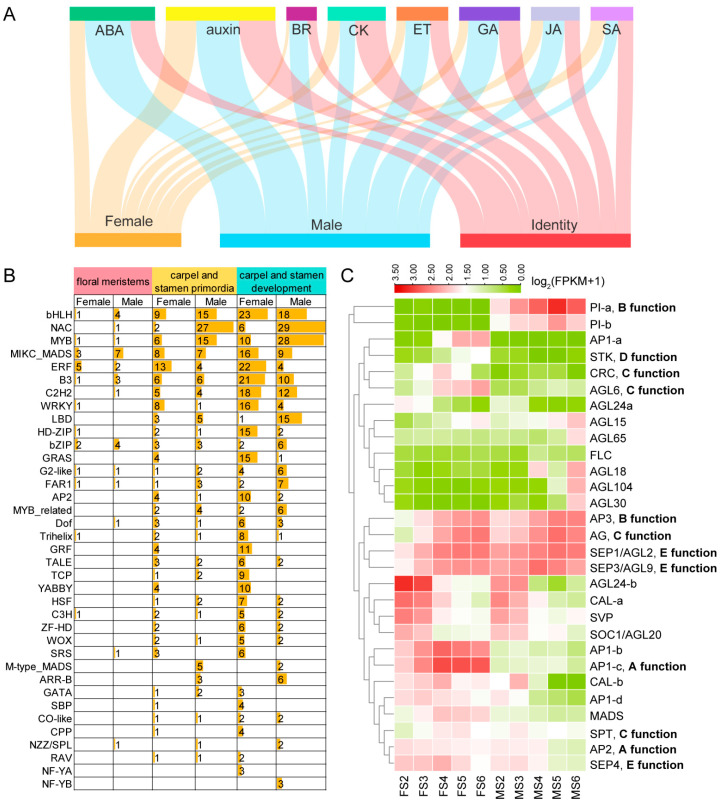
Analysis and comparison of DEGs in female and male development. (**A**) Number of DEGs involved in phytohormone pathways in flower development. (**B**) Statistics of TF family members as DEGs during female and male development. (**C**) Gene expression patterns of MIKCc and ABCDE models of various organs of spinach. Expression values were scaled by log2(FPKM + 1), in which FPKM is fragments per kilobase of exon per million mapped reads. F: female; M: male; S: stage.

**Figure 3 ijms-25-06127-f003:**
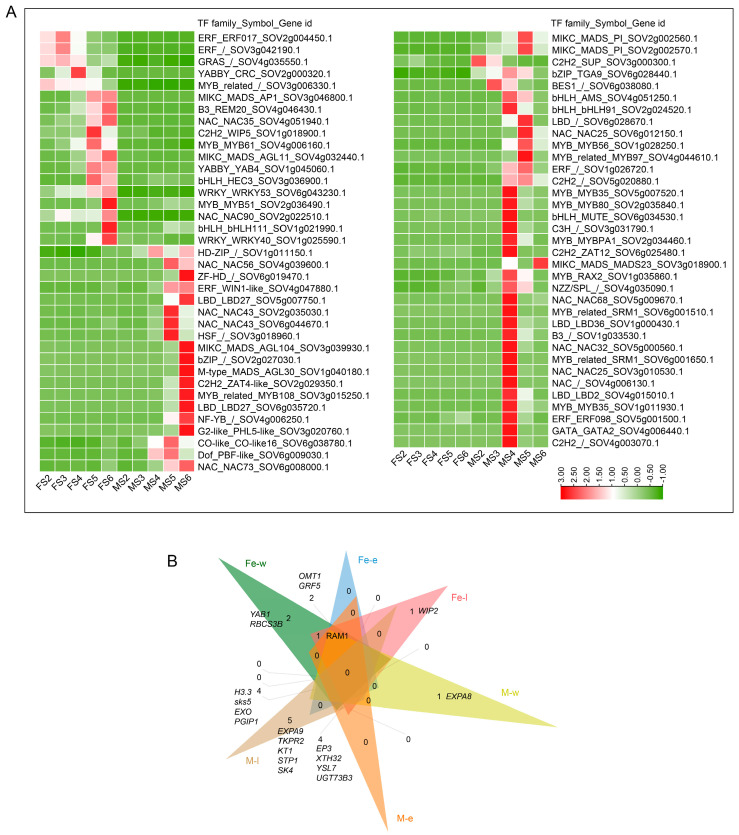
(**A**) Specifically expressed TFs in dioecious spinach female and male flower development. F: female; M: male; S: stage. (**B**) Target high-frequency DEGs regulated by floral regulators. w: the whole stage of flower development; e: the early stage of flower development; l: the late stage of flower development.

**Figure 4 ijms-25-06127-f004:**
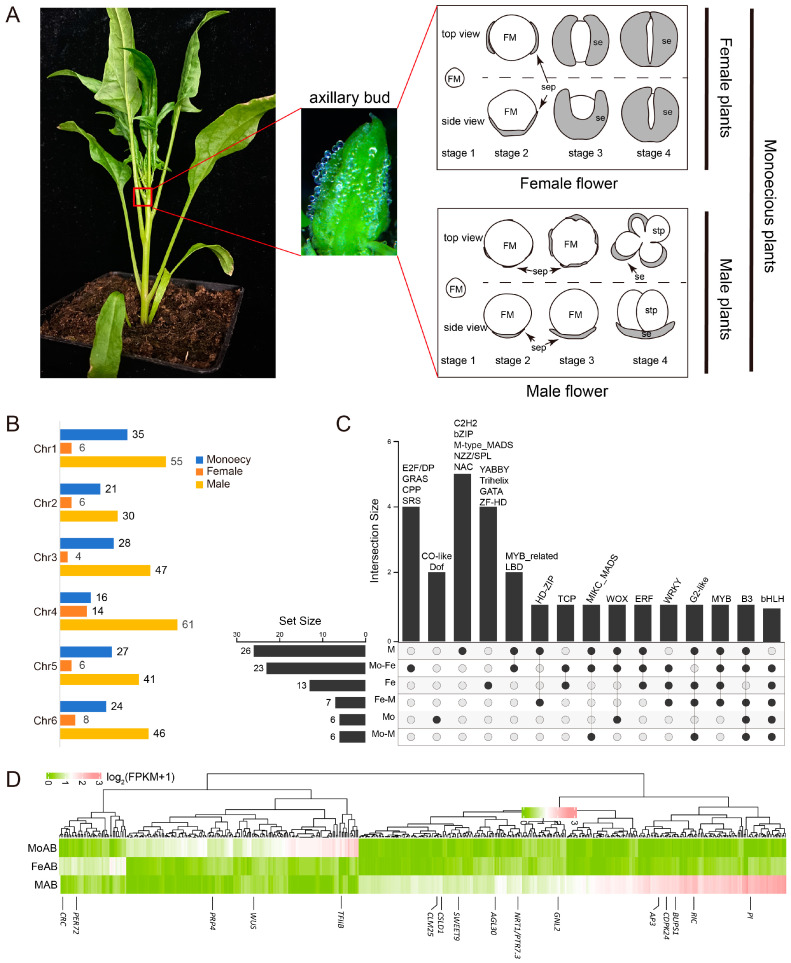
Transcriptomic profiles during the early-stage development of female, male, and monoecious axillary buds. (**A**) The morphology of flowers during early-stage development in female, male, and monoecious plants. (**B**) The chromosomal distributions of sex-biased genes in female, male, and monoecious axillary buds. (**C**) Number of sex-biased transcription factors. Fe: female; M: male; Mo: monoecious. (**D**) Expression profiles of specific DEGs of female, male, and monoecious. Expression values were scaled by log2(FPKM + 1), in which FPKM is the fragments per kilobase of exon per million mapped reads. AB: axillary bud.

**Figure 5 ijms-25-06127-f005:**
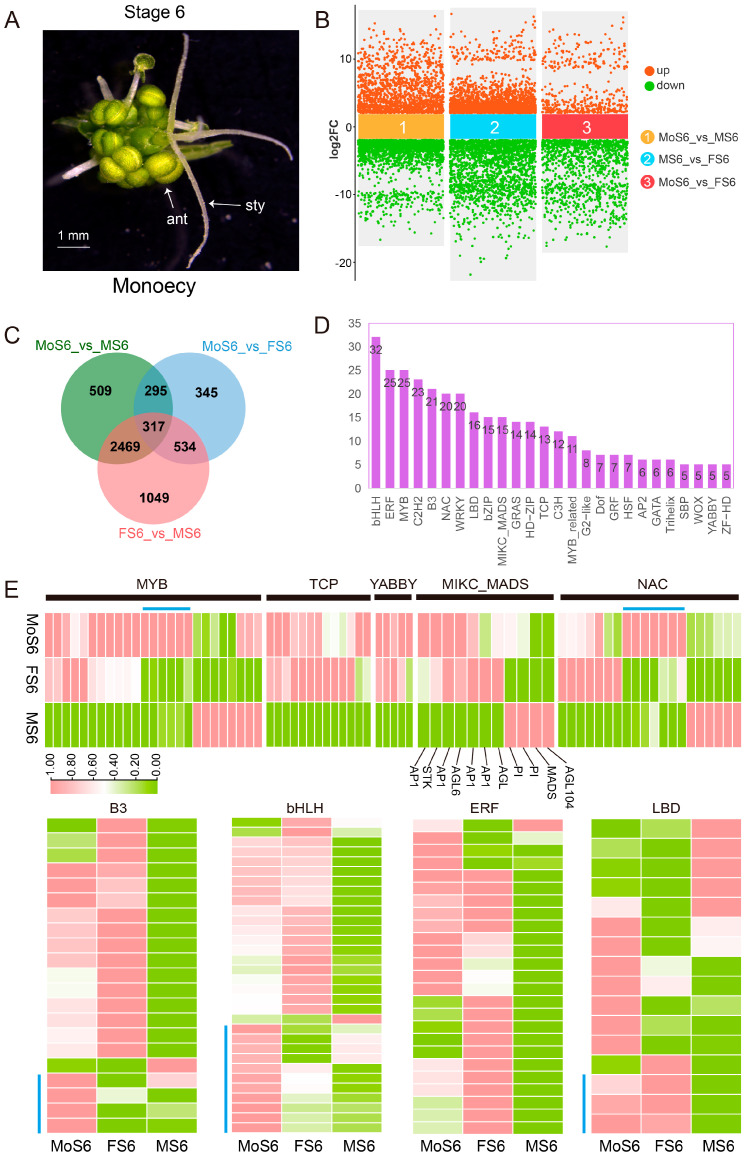
Transcriptomic profiles during the mature stage development of female, male, and monoecious flowers in spinach. (**A**) The morphology of spinach monoecious flowers in the mature stage. (**B**,**C**) DEGs among female, male, and monoecious flowers. F: female; M: male; Mo: monoecious; S: stage. (**D**) Number of differentially expressed transcription factors. (**E**) Expression profiles of major transcription factor families. Blue line: Genes significantly expressed in monoecious mature flowers.

**Figure 6 ijms-25-06127-f006:**
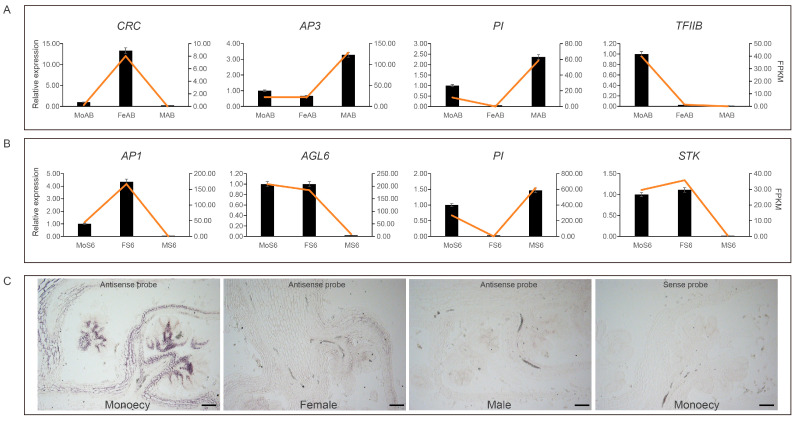
Validation of selected genes expression. (**A**,**B**) Relative expression levels of selected genes from RT-qPCR analysis. (**C**) *SpTFIIB* expression patterns in axillary buds of male, female, and monoecious flowers. In situ hybridization was performed with antisense and sense *SpTFIIB* RNA probes. Bar = 100 μm.

**Figure 7 ijms-25-06127-f007:**
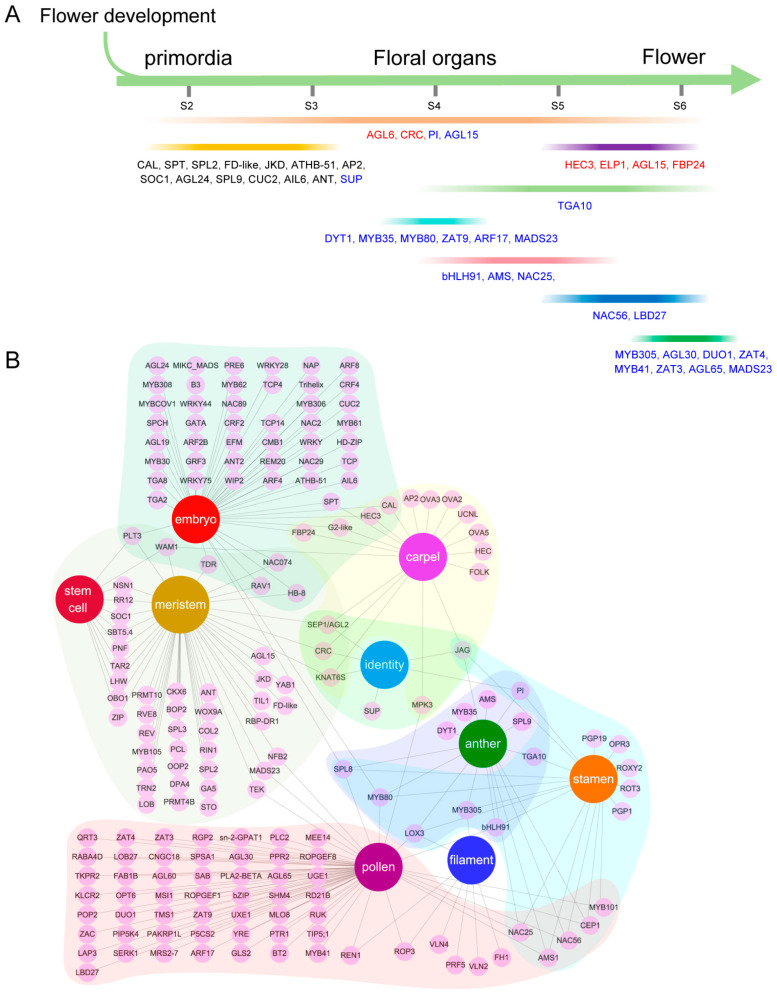
Key regulatory TFs throughout spinach flower development. (**A**) Schema depicting the key regulatory transcription factors in the flower developmental stages of flower formation, including floral primordia (S2), organ specification and differentiation (S3–4), and flower maturation (S5–6). S: stage; Red font: regulatory female TFs; blue font: regulatory male TFs. (**B**) Network showing organ or tissue-specific TFs during spinach flower development.

**Figure 8 ijms-25-06127-f008:**
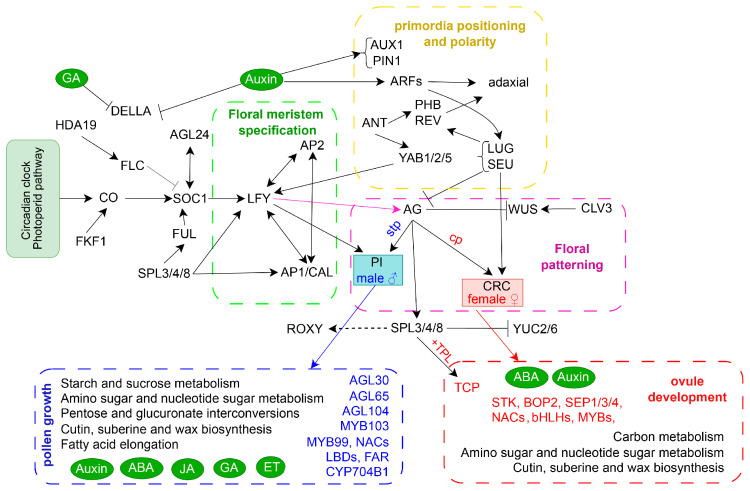
Schematic diagram of the regulatory mechanism for floral development in dioecious spinach. stp: stamen primordia; cp: carpel primordia.

## Data Availability

The transcriptome sequencing data of monoecious plants were submitted to the National Center for Biotechnology Information Sequence Read Archive (SRA) under Accession Number PRJNA1062132. Transcriptome data of dioecious female and male flowers are under PRJNA649901 and PRJNA724923, respectively.

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
