# Peer review of "Gene Regulatory Network Controlling Flower Development in Spinach (Spinacia oleracea L.)"

_ijms, 2024, doi:10.3390/ijms25116127_

Round 1

Reviewer 1 Report

Comments and Suggestions for Authors

Summary Comments:

The paper provides a comprehensive analysis of spinach flower development, covering various stages from organ identity determination to maturation. It employs RNA-seq data to investigate gene expression patterns and regulatory mechanisms associated with different aspects of flower development in dioecious spinach. The paper presents findings in a clear and structured manner, facilitating understanding for readers. Overall, the paper offers valuable insights into spinach flower development and provides a solid foundation for further research in this field. 

Areas for Improvement:

1> Some sections could benefit from clearer organization and subheadings to improve readability and navigation. While supplementary data are referenced, including more detailed tables or figures within the main text would enhance the presentation of key findings.

2> The methods section could be more detailed and transparent, particularly regarding the experimental procedures for RNA-seq analysis, RT-qPCR validation, and in situ hybridization experiments.

3> While RT-qPCR and in situ hybridization are valuable validation techniques, the paper needs to select more genes to validate the transcriptome results. Also paper could benefit from additional validation methods to corroborate the RNA-seq findings by including alternative validation approaches, such as Western blotting or immunohistochemistry, would strengthen the robustness of the results.

4> The paper should provide information on sample size and replication for both RNA-seq and validation experiments.

5> While the descriptive results offer valuable information about gene expression patterns and hormonal regulation during flower development in spinach, deeper insights could be gained by exploring the functional significance of these expression patterns, elucidating regulatory networks, and investigating how specific genes and pathways contribute to flower development and sexual differentiation.

Reviewer 2 Report

Comments and Suggestions for Authors

I read with interest the manuscript entitled “Gene regulatory network controlling flower development in spinach”. This study aimed to understand the differences in sexual differentiation and floral organ development of dioecious flowers, as well as the differences in the regulatory mechanisms of floral organ development of dioecious and monoecious flowers. Therefore, the manuscript needs some adjustments so that it can then be forwarded to the publication process. The manuscript has the potential for publication in this journal International Journal of Molecular Sciences and needs the following adjustments:

TITLE

Add the scientific name of the species. The common name can change throughout the world.

ABSTRACT

Replace the keywords that are repeated in the title. To improve the search for future work to be published, the words mentioned in the title must be different from the Keywords.

INTRODUCTION

Add hypotheses about the work. This information must be before the objectives.

RESULTS

- What program was used to make the figures? This should be mentioned in the material and methods section.

- Are all these files necessary as supplementary documents? Check and leave only what is necessary for the research.

MATERIAL AND METHODS

- What was the location of the study? Mention and cite geographic coordinates. This information about the study location is important for other researchers interested in the research.

- What was the statistical analysis used in this study?

- I suggest adding a subtopic with this information. There is nothing described about this.

-

CONCLUSION

- To review. The initial information has already been mentioned in the objectives.

- No need to cite Table or Figure. Delete.

- Review this section in a general way.

Reviewer 3 Report

Comments and Suggestions for Authors

The study provides transcriptomic analysis to compare differences between dioecious and monoecious flowers, highlighting the involvement of key transcription factor families and hormonal pathways. The manuscript is clear and well-written that provides significant mechanisms on understanding of spinach flower development.

However, the following comments should be taken into consideration for further improvement:

  1. While the manuscript is generally well-organized, I recommend revising in introduction part for clarity. Specifically, providing a clearer delineation of the research questions/hypotheses and a more structured presentation of the methodology used for transcriptomic analysis and DEG identification would enhance readability.
  2. If feasible please convert the Table 1 as a figure.
  3. In the Discussion section please provide additional context on how the study's findings contribute to existing knowledge in the field of plant developmental biology would strengthen the manuscript's significance and relevance. https://doi.org/10.1186/s12870-021-02956-0

4.     Please specify the number of reference genes utilized for calculating the gene expression of the targeted gene.

5.     Please revise the conclusion part.

Overall, the manuscript presents valuable findings on spinach floral development. Addressing the minor revisions outlined above would further improve clarity, organization, and contextualization of the results.

Round 2

Reviewer 2 Report

Comments and Suggestions for Authors

The article has been corrected, as per previous suggestions.